# Paratransgenic manipulation of a tsetse microRNA alters the physiological homeostasis of the fly's midgut environment

Liu Yang[1]*, Brian L. Weiss[1]*, Adeline E. Williams[1,2], Emre Aksoy[1,3], Alessandra de Silva Orfano[1], Jae Hak Son[1], Yineng Wu[1], Aurelien Vigneron[1,4], Mehmet Karakus[1,5], Serap Aksoy[1]

1 Department of Epidemiology of Microbial Diseases, Yale School of Public Health, New Haven, Connecticut, United States of America, 2 Department of Microbiology, Immunology, Pathology, Colorado State University, Fort Collins, Colorado, United States of America, 3 Department of Immunology and Infectious Diseases, Harvard T.H. Chan School of Public Health, Boston, Massachusetts, United States of America, 4 Department of Evolutionary Ecology, Institute for Organismic and Molecular Evolution, Johannes Gutenberg University, Mainz, Germany, 5 Department of Medical Microbiology, Faculty of Medicine, University of Health Sciences, Istanbul, Turkey

* liu.yang.ly274@yale.edu (LY); brian.weiss@yale.edu (BLW)

**Data Availability Statement:** The data is now being held at NCBI SRA. Bioproject ID:

## Abstract

Tsetse flies are vectors of parasitic African trypanosomes, the etiological agents of human and animal African trypanosomoses. Current disease control methods include fly-repelling pesticides, fly trapping, and chemotherapeutic treatment of infected people and animals. Inhibiting tsetse's ability to transmit trypanosomes by strengthening the fly's natural barriers can serve as an alternative approach to reduce disease. The peritrophic matrix (PM) is a chitinous and proteinaceous barrier that lines the insect midgut and serves as a protective barrier that inhibits infection with pathogens. African trypanosomes must cross tsetse's PM in order to establish an infection in the fly, and PM structural integrity negatively correlates with trypanosome infection outcomes. Bloodstream form trypanosomes shed variant surface glycoproteins (VSG) into tsetse's gut lumen early during the infection establishment, and free VSG molecules are internalized by the fly's PM-producing cardia. This process results in a reduction in the expression of a tsetse microRNA (*miR275*) and a sequential molecular cascade that compromises PM integrity. miRNAs are small non-coding RNAs that are critical in regulating many physiological processes. In the present study, we investigated the role(s) of tsetse *miR275* by developing a paratransgenic expression system that employs tsetse's facultative bacterial endosymbiont, *Sodalis glossinidius*, to express tandem antagomir-*275* repeats (or *miR275* sponges). This system induces a constitutive, 40% reduction in *miR275* transcript abundance in the fly's midgut and results in obstructed blood digestion (gut weights increased by 52%), a significant increase (p-value < 0.0001) in fly survival following infection with an entomopathogenic bacteria, and a 78% increase in trypanosome infection prevalence. RNA sequencing of cardia and midgut tissues from paratransgenic tsetse confirmed that *miR275* regulates processes related to the expression of PM-associated proteins and digestive enzymes as well as genes that encode abundant secretory proteins. Our

PRJNA732457. https://www.ncbi.nlm.nih.gov/sra/
PRJNA732457.

**Funding:** Funding was provided by NIH/NIAID
(R01AI139525), the Li Foundation and Ambrose
Monell Foundation to SA. The funders had no role
in study design, data collection and analysis,
decision to publish, or preparation of the
manuscript.

**Competing interests:** The authors have declared
that no competing interests exist.

study demonstrates that paratransgenesis can be employed to study microRNA regulated pathways in arthropods that house symbiotic bacteria.

## Author summary

Tsetse flies transmit African trypanosomes, which are the parasites that cause sleeping sickness in human in sub-Saharan Africa. When tsetse ingests a blood meal containing trypanosomes, the expression level of a microRNA (*miR275*) decreases in the fly's gut. This process results in a series of events that interrupt the physiological homeostasis of the gut environment. To further understand the function of *miR275* in tsetse fly, we genetically modified a tsetse's native bacterial symbiont, reintroduced the genetically modified bacterium back into the fly, and successfully knocked down the *miR275* expression in tsetse's midgut. These 'paratransgenic' flies (which house genetically modified bacteria) presented impaired digestive processes and were highly susceptible to infection with trypanosomes. Lastly, we discovered that *miR275* regulates tsetse digestive processes and secretory pathways. Our novel paratransgenic expression system can be applied to study the function of other microRNAs and how they regulate disease transmission in tsetse and other insect systems.

## Introduction

Tsetse flies (*Glossina* spp.) are obligate vectors of pathogenic African trypanosomes throughout 37 countries in sub-Saharan Africa [1]. These parasites belong to the genus of *Trypanosoma*, which includes the important human pathogens *T. b. rhodesiense* and *T. b. gambiense* (the causative agents of human African trypanosomiasis, or HAT) and non-human animal pathogens *T. b. brucei*, *T. congolense* and *T. vivax* (the causative agents of animal African trypanosomiasis, or AAT). Both AAT and HAT are fatal if left untreated [2]. Current disease control methods include vector control to reduce population size and chemotherapeutic treatment of infected people and domesticated animals [3]. A more complete molecular understanding of tsetse-trypanosome interactions will facilitate the development of novel control strategies, such as reducing or eliminating the fly's capacity to transmit trypanosomes.

The tsetse-specific stages of the trypanosome life cycle begin when the fly ingests a bloodmeal that contains mammalian stage bloodstream form (BSF) parasites. Upon ingestion by tsetse, BSF parasites differentiate into insect adapted procyclic forms (PCF) in the lumen of the fly's midgut [4,5]. PCF parasites then bypass the fly's peritrophic matrix (PM) barrier in the anterior midgut and replicate within the ectoperitrophic space (ES, the region between the PM and the midgut epithelia) [6–8]. As part of their development from BSF to PCF parasites, the BSF trypanosomes shed their abundant surface coat antigens, known as variant surface glycoprotein (VSG) into the fly's midgut lumen. Free VSG is transiently internalized by cells of tsetse's PM-producing cardia (also known as proventriculus) [9,10]. This process reduces the expression of genes that encode PM associated proteins and digestive enzymes, and modulates the expression of several microRNAs, including a drastic reduction in the expression of tsetse *microRNA 275* (*miR275*) [10].

miRNAs are small (~23 nucleotides) non-coding RNAs that regulate many important physiological processes. miRNAs often suppress gene expression by guiding the Argonaute (AGO) protein to bind with its target mRNA, which induces the miRNA induced silencing complex

(miRISC) and leads to post-transcriptional repression or degradation of the target mRNA [11–13]. miRNAs can also upregulate gene expression by inducing translational activation [14,15]. When the expression of *miR275* was experimentally reduced in tsetse's cardia and midgut through the provisioning of synthetic anti-*miR275* antagomirs (antagomir-275) or VSG purified from BSF trypanosomes, formation of the fly's PM was impaired. This process disrupted blood meal digestion and enhanced the ability of trypanosomes to establish an infection in the fly's midgut [10]. In the mosquito *Aedes aegypti*, *miR275* similarly influences midgut blood digestion and fluid excretion by regulating the expression of its target gene *SERCA* (sarco/endoplasmic reticulum Ca2+ adenosine triphosphatase) [16,17], but the mRNA target of *miR275* in tsetse remains unknown.

Tsetse flies house a consortium of symbiotic microbes that mediate numerous aspects of their host's physiology [18,19]. One of these is the facultative endosymbiotic bacterium *Sodalis glossinidius*, which resides extra- and intracellularly within multiple tsetse tissues, including the midgut, salivary glands, and reproductive organs [20]. *Sodalis* can be cultivated and genetically modified *in vitro*, and recolonized into tsetse's gut via a blood meal [21,22]. Reintroducing recombinant *Sodalis* (rec*Sodalis*) does not elicit immune responses that would induce any fitness cost [22,23]. *Per os* provisioned rec*Sodalis* remains only in the gut [22]. 'Paratransgenic' tsetse flies that house rec*Sodalis* have been successfully used to deliver anti-trypanosomal nanobodies [24–26]. Paratransgenesis has also been used to deliver dsRNA for gene silencing in kissing bugs [27,28] and in the malaria mosquito *Anopheles gambiae* [29,30], and to express intronic miRNA (using a recombinant densovirus) in the mosquito *Aedes albopictus* [31]. However, paratransgenic expression of small RNA antagomirs to knockdown miRNA expression in tsetse fly has not been reported to date. Herein we engineered *Sodalis* to paratransgenically express three tandem antagomir-275 repeats (3xant-*miR275*) in tsetse's cardia and midgut environments, and then used this experimental system to investigate the mechanism(s) by which *miR275* regulates the physiological homeostasis of the fly's gut environment. We found that paratransgenic flies presented multiple phenotypes that are associated with the production of a structurally compromised PM barrier and/or disrupted gut homeostasis. Our novel paratransgenic expression system can be applied to further study functions of microRNAs that are involved in the tsetse-trypanosome interaction, thus advancing our understanding of parasite-deployed strategies to manipulates its host physiology. Additionally, this method could be broadly applied to other arthropod systems where a host interacts with microbes (especially with non-model systems where host genetic manipulation can be difficult), which could be particularly useful to study pathogen-host interactions in the field of vector biology.

## Materials and methods

### Tsetse fly and bacterial cultures

Tsetse flies (*Glossina morsitans morsitans*, *Gmm*) were reared in the Yale University insectary at 25°C and 70% relative humidity (RH), and received defibrinated bovine blood every 48 h via an artificial blood feeding system. Wild-type *Sodalis glossinidius morsitans* were isolated from surface-sterilized *Gmm* pupae and plated on Difco Brain Heart Infusion Agar (BD Biosciences) plates that were supplemented with 10% bovine blood (BBHI). Clonal *Sodalis* populations were subsequently maintained *in vitro* in Bacto Brain Heart Infusion (BHI) medium (BD biosciences) at 26°C, 10% $CO_2$.

### Generation of recSodalis strains

To generate rec*Sodalis*, two constructs (Fig 1A) were made using a modified pgRNA-bacteria plasmid (NEB, Addgene plasmid # 44251). This plasmid, which encodes an ampicillin

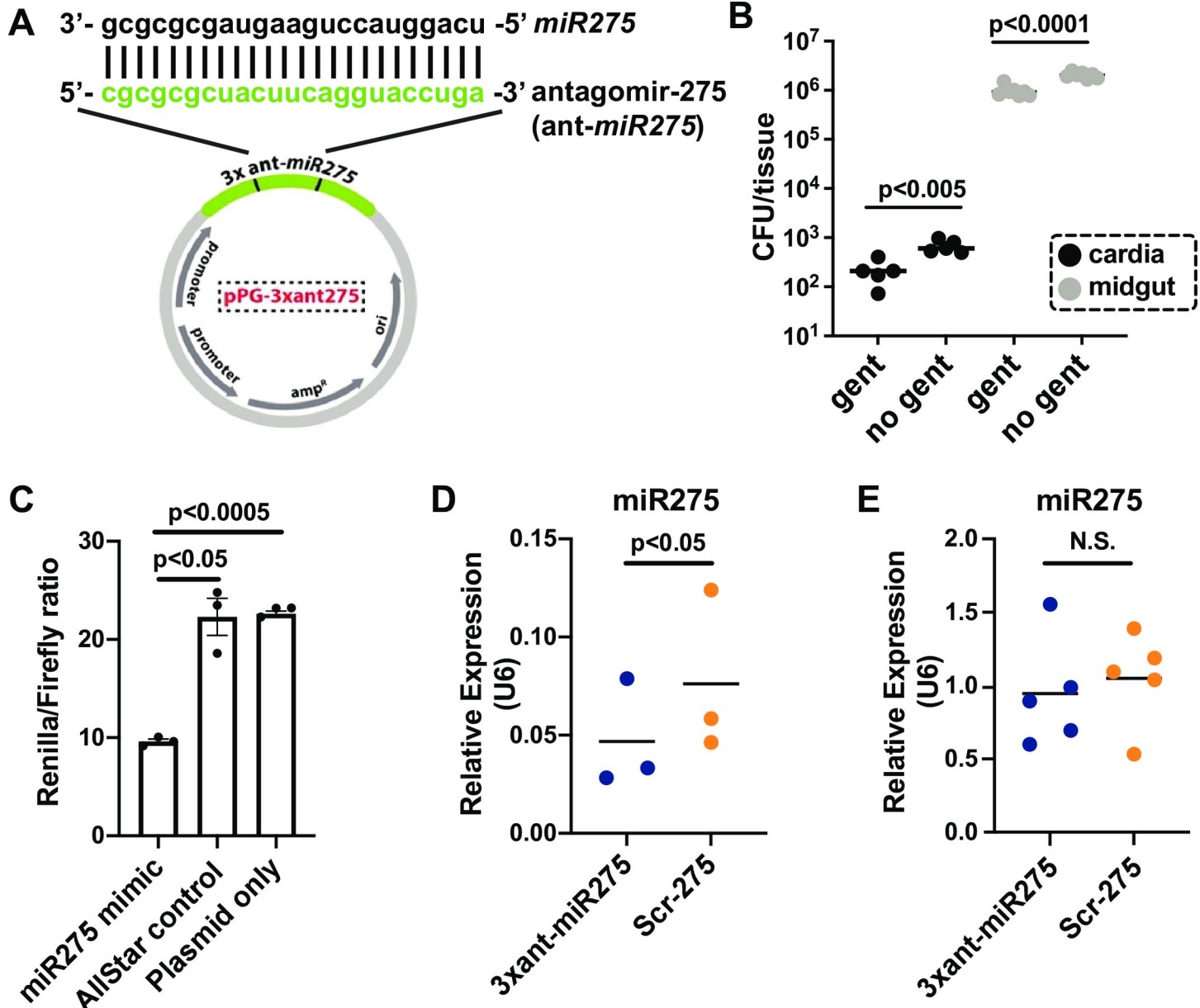

**Fig 1. The successful development of paratransgenic expression system.** (A) rec*Sodalis* plasmid construct. Three tandem antagomir-275 repeats (3xant-*miR275*, in green) that are complementary to the tsetse *miR275* mature sequence were cloned into plasmid pgRNA. Each repeat is separated by a 3-nucleotide linker sequence. 3xant-*miR275*, and a similarly engineered construct that encodes a scrambled antagomir-275 (Scr-*275*), were electroporated into *Sodalis*^WT to generate strains designated *Sgm*^3xant-*miR275* and *Sgm*^Scr-*275*, respectively. (B) Quantification of *Sgm*^3xant-*miR275* within cells of tsetse's cardia (black) and midgut (grey) via gentamicin exclusion assay. Each dot represents one tsetse organ (*n* = 5). A Student's t-test was used to determine statistical significance. (C) Dual luciferase reporter assay. Each dot represents the average of normalized luciferase signal (*Renilla*/*Firefly* ratio) ± SEM of each experiment. The 3xant-*miR275* construct was cloned into the psiCheck-2 plasmid containing two luciferase reporter genes, *Renilla* (reporter) and *Firefly* (internal control). The luciferase activity is measured by the *Renilla* signal normalized to the *Firefly* signal. Three different experiments were performed to test the binding efficacy between 3xant-*miR275* and 1) synthetic miR275 mimic, 2) synthetic AllStars Negative Control, and 3) psiCheck plasmid without adding any miRNA. Three biological replicates (with 3 technical replicates each) per experiment were used. Bonferroni's multiple comparison tests were used to determine statistical significance. (D) *miR275* expression level in the midgut of paratransgenic *Gmm*^3xant-*miR275* versus *Gmm*^Scr-*275* flies. Each dot represents 5 individual midguts. A student's t-test was used to determine statistical significance. (E) *miR275* expression in the cardia of *Gmm*^3xant-*miR275* versus *Gmm*^Scr-*275* flies. Each dot represents 5 individual cardia. A student's t-test was used for statistical analysis.

**Table 1. Oligonucleotide sequences.** Capitalized letters represent restriction endonuclease cut sites. Red = antagomir-275.

| Name | Strand | Sequence |
|---|---|---|
| 3xant-*miR275* | F | CTAGTcgcgcgctacttcaggtacctgaatccgcgcgcgctacttcaggtacctgaatccgcgcgcgctacttcaggtacctgaCCTGCAGGtcaacttgaaaaagtggcaccgagtcggtgctttttttga |
| | R | AGCTtcaaaaaaagcaccgactcggtgccactttttcaagttgaCCTGCAGGtcaggtacctgaagtagcgcgcggattcaggtacctgaagtagcgcgcggattcaggtacctgaagtagcgcgcgA |
| Scr-275 | F | CTAGTaccggcttagtaagaggctagttagcatcacgtcttccattttgctcaatggcataggatgtcgttcgttggcgtgtcgggacctcgcaagagattaaCCTGCA |
| | R | GGttaatctcttgcgaggtcccgacacgccaacgaacgacatcctatgccattgagcaaaatggaagacgtgatgctaactagcctcttactaagccggtA |

resistance cassette, was originally designed to express short guide RNAs for CRISPR application and is thus well suited for expressing small RNAs [32]. An additional endonuclease cut site Sbfi was built into the original pgRNA plasmid backbone so as to include an RNA terminator sequence in the modified plasmid. Two pairs of two complementary single-stranded oligonucleotides (oligos) that encode either three copies of the *miR275* antagomir (3xant-*miR275*) or the scrambled *miR275* control (Scr-275) were synthesized at Yale Keck Oligo Synthesis Resource (Table 1). The two antisense single-stranded oligos each contains a bulge (not perfectly complementary) at the linker sequence to increase the effectivity of the sponge construct [33]. Each strand encodes SpeI and Sbfi restriction endonuclease cut sites, were annealed at 95˚C for 5 min, cooled to room temperature for 30 min and stored at -20˚C for future use. Both pgRNA and the double stranded miRNA-encoding oligos were subjected to restriction endonuclease treatment by SpeI and Sbfi at 37˚C for 2 h. The oligos were then ligated into pgRNA using T4 DNA ligase (NEB), and the constructs were propagated in *E. coli DH5a* cells. All purified plasmid constructs were sequenced at Yale's Keck Sequencing Laboratory to confirm their structure.

The purified DNA plasmids were electroporated into wild-type *S. glossinidius morsitans* ($Sgm^{WT}$) as described previously [34]. Two rec*Sodalis* strains were used in this study: 1) $Sgm^{3xant-miR275}$, which encodes 3xant-*miR275*, and 2) the *miR275* scrambled control ($Sgm^{Scr-275}$) (Table 1). In brief, 25 mL of log-phase *Sodalis* cells ($OD_{600} = 0.3~0.5$; SmartSpec Plus spectrophotometer; Bio-Rad, Hercules, CA) were washed consecutively in 25 mL, 1 mL and 1 mL 10% sterile pre-chilled glycerol. After the three washes, the *Sodalis* cell pellets were resuspended in 50 μL sterile 10% glycerol. Each 50 μL of cell mixture was mixed with 1 or 2 μL (~100 ng) of plasmid DNA and subjected to electroporation (voltage, 1.9 kV; capacitance, 25 uF; resistance, 200 omega). After electroporation, the rec*Sodalis* cells were immediately placed in 5 mL BHI medium for overnight recovery at 26˚C, 10% $CO_2$. The recovered cells were then plated on BHI plates supplemented with 10% bovine blood, and transformants were selected with ampicillin (50 μg/mL). After a 1-week incubation, transformants were selected for PCR and sequencing. After the sequence was confirmed, a single recS*odalis* colony was grown in BHI medium for future experiments.

## Establishment of paratransgenic tsetse flies

To generate paratransgenic tsetse flies, two groups of teneral female flies (newly emerged unfed adults) were given two consecutive blood meals (separated by 1 day) containing either $Sgm^{3xant-miR275}$ or $Sgm^{Scr-275}$ ($10^6$ CFU/mL each in the first two blood meals) and ampicillin (50 μg/mL). After a third blood meal (no rec*Sodalis*, no ampicillin), 8-day old paratransgenic flies were used in the experiments described below.

## Gentamicin exclusion assay and quantification of recSodalis

Gentamicin is unable to cross the eukaryotic cell membrane and hence only kills extracellular bacteria [35]. Cardia and midgut tissues were dissected from 8-day old paratransgenic and

incubated in sterile 0.85% NaCl supplemented with 100 μg/mL gentamicin. Controls were incubated in the sterile NaCl in the absence of gentamicin. Tissues were agitated on a shaking platform at room temperature for 1 h and washed 4 times in 500 μl sterile 0.85% NaCl. After the 4th wash, tissues were rigorously homogenized in sterile 0.85% NaCl. 50 μl of lysate from each treatment was plated onto BHI Agar plates supplemented with 10% blood and 50 μg/mL ampicillin. After 7 days of incubation at 26˚C, 10% $CO_2$, colonies on each plate were counted as described in [22]. Multiple colonies were randomly selected for colony PCR (with primers targeting the inserted section of the pgRNA plasmid) and subjected to sequencing to confirm they housed the correct plasmid construct.

## Dual luciferase reporter assay

To clone the 3xant-*miR-275* into psiCheck-2 (Promega), two complementary single-stranded oligos that encode 3xant-*miR-275* and XhoI and NotI restriction endonuclease cut sites were synthesized at Yale Keck Oligo Synthesis Resource (Table 1). The complementary oligos were annealed at 95˚C for 4 min and cooled to room temperature for 30 min. The psiCheck-2 vector and the doubled stranded miRNA-encoding oligos were subjected to XhoI and NotI treatment at 37˚C for 2 h followed by inactivation at 65˚C. The oligos were then ligated into the double digested psiCheck-2 plasmid using T4 DNA ligase (NEB) at room temperature for 2 h, and the constructs were propagated in *E. coli DH5a* cells. All purified plasmid constructs were sequenced at Yale's Keck Sequencing Laboratory to confirm their structure. The psiCheck-2 vector containing the 3xant-*miR275* sequence is hereafter referred to as psiCheck-2$^{3xant-miR275}$.

For transfection, *Drosophila* S2 cells (Invitrogen) were maintained at 28˚C in Schneider *Drosophila* medium supplemented with 10% heat inactivated FBS. We co-transfected 100 ng of psiCheck-2$^{3xant-miR275}$ and the synthetic tsetse *miR275*miScript miRNA mimic at 100 nM (Qiagen) or with AllStars Negative Control (Qiagen) into S2 cell lines using Attractene Transfection reagent following the manufacturer's protocol (Qiagen). A "no miRNA" treatment with only psiCheck-2 plasmid and transfection reagent was also conducted. Dual luciferase reporter assays were performed 48 h post transfection using the Dual Luciferase Reporter Assay System following the manufacturer's protocol (Promega). The *renilla* (primary reporter) luciferase signal was normalized to the *firefly* (internal control) luciferase signal. Each treatment was conducted triplicate.

## Quantitative real-time PCR

Quantitative real-time PCR (qPCR) was used to quantify the expression levels of *miR275*, non-coding small nuclear RNA (snRNA) *U6*, and saliva-associated genes in our paratransgenic flies (described in section 2.3 above). Tsetse cardia, midgut and salivary glands were microscopically dissected 24–48 h after the third blood meal. Total RNA was extracted from pools of 5 cardia, 5 midgut or 10 salivary glands (as one biological replicate) using Trizol reagent [36]. RNA was cleaned and purified using an RNA Clean and Concentrator Kit with in-column DNase treatment (Zymo Research). RNA quality and quantity was quantified using a NanoDrop 2000c (Thermo Scientific). 100 ng of cardia, 500 ng of midgut or 100 ng of salivary glands RNA, respectively, was then reverse transcribed into cDNA using the miScript II RT kit (Qiagen 218160) followed by qPCR. For each sample, two technical replicates were used. Relative expression (RE) was measured as RE = 2$^{-ddCT}$, and normalization was performed using *U6* gene expression as a reference. Primers for amplifying *miR275*, saliva-associated genes and the reference gene are listed in S1 Table.

qPCR was performed on a CFX96 PCR detection system (Bio-Rad, Hercules, CA) under the following conditions: 8 min at 95˚C; 40 cycles of 15 s at 95 ˚C, 30 s at 57 ˚C or 55 ˚C, 30 s at 72 ˚C; 1 min at 95 ˚C; 1 min at 55 ˚C and 30 s from 55 ˚C to 95 ˚C. Each reaction consisted of

10 μl: 5 μl of iTaq$^{TM}$ Universal SYBR Green Supermix (Bio-Rad), 1 μl cDNA, 2 μl primer pair mix (10 μM) and 2 μl nuclease-free $H_2O$.

### Tsetse whole gut weight measurements

Individual guts from 8-day old paratransgenic flies ($n$ = 20 per group) were dissected 24 h after their last blood meal and weighed with a digital scale as an indicator for blood digestion.

### Serratia infection assay

8-day old paratransgenic individuals were fed a blood meal containing $10^3$ CFU/mL *S. marcescens* strain Db11. Thereafter, all flies were maintained on normal blood and their mortality was recorded every other day for 14 days. Details of the *Serratia* infection assay are provided in [6,9,10] as well as in the discussion section.

### Trypanosome infection prevalence

The 8-day old paratransgenic flies were challenged *per os* with a blood meal containing $10^7$ CFU/mL *Trypanosoma brucei brucei* strain 503 supplemented with 0.9 mg/mL of cysteine. Thereafter, the flies were maintained on normal blood meals for two weeks. Their guts were dissected and microscopically examined to determine trypanosome infection status.

### mRNA library construction and RNA sequencing

Two groups of paratransgenic flies ($Gmm^{3xant-miR275}$ vs. $Gmm^{Scr-275}$) were generated as described in Section 2.3. All flies were dissected 36 h after the third blood meal; 10 individual cardia or 5 individual midgut were pooled as one biological replicate and stored in -80˚C prior to RNA extraction, a total 3 biological replicates per treatment were used. Total RNA was extracted using Trizol reagent according to the manufacturer's protocol (Invitrogen), followed by RNA Clean and Concentrator Kit and in-column DNase treatment (Zymo Research). RNA quality and quantity were quantified using a bioanalyzer. All 12 mRNA libraries were prepared and sequenced (pair-ended) at Yale Center for Genome Analysis (YCGA) using Illumina NovaSeq system.

### RNA-seq data processing

RNA-seq raw reads were uploaded to FastQC (v. 0.11.9, www.bioinformatics.babraham.ac.uk/projects/) for quality check, and then trimmed and filtered to remove ambiguous nucleotides and low-quality sequences. The reads were mapped to *Glossina morsitans morsitans* reference genome [37] using HISAT2 v2.1.0 with default parameters [38,39]. We then used the function 'htseq-count' in HTSeq v0.11.2 [40] to count the number of reads mapped to the genes annotated in the reference genome (version GmorY1.9 at Vectorbase.org) with option "-s reverse". Reads that were uniquely aligned to *Gmm* transcripts were used to calculate differential gene expression using *EdgeR* package in R software [41]. Significance was determined using EdgeR General linear models, corrected with a False Discovery Rate (FDR) at $p < 0.05$. The differentially expressed (DE) genes were uploaded to VectorBase (http://beta.vectorbase.org) for gene ontology (GO) enrichment analysis using the built-in web tool GO Enrichment analysis. REVIGO was used to remove the redundant GO terms [42].

### Replicates and statistics

Biological replicates were obtained from samples derived from distinctly repeated experiments. Details about sample sizes and statistical tests used for data analyses in this study are indicated in the corresponding figure legends.

## Results

### Successfully developed the paratransgenic expression system

To knock down expression of tsetse *miR275*, we designed two expression constructs that encode 1) 3xant-*miR275* to knockdown *miR275*, and 2) a scrambled miR275 sequence (Scr-275) that served as the control. Individual clonal populations of wild-type *Sodalis* (*Sgm*^WT) were transformed with one of the plasmids and are henceforth designated *Sgm*^3xant-*miR275* and *Sgm*^Scr-275 (Fig 1A). We then colonized individual groups of newly emerged (teneral) adult tsetse *per os* with either *Sgm*^3xant-*miR275* or *Sgm*^Scr-275, thus generating paratransgenic tsetse cohorts designated *Gmm*^3xant-*miR275* (treatment) and *Gmm*^Scr-275 (control), respectively. During the development of the paratransgenic lines, we supplemented the first two bloodmeals with ampicillin to suppress the *Sgm*^WT population, which provided the antibiotic-resistant rec*Sodalis* populations a selective advantage over the indigenous antibiotic susceptible WT cells.

We performed gentamicin exclusion assays to confirm that the rec*Sodalis* successfully invaded tsetse cardia and midgut cells. Gentamicin cannot penetrate eukaryotic cell membranes, and thus treatment with this antibiotic effectively eliminates the extracellular bacteria but leaves the intracellular population intact [35]. We incubated separate cardia and midgut tissues dissected from 8-day old paratransgenic flies in either gentamicin (treatment) or PBS (control). Tissues were subsequently rinsed, homogenized, and plated on BBHI plates supplemented with ampicillin. We recovered 214 ($\pm$ 54.0) and $9.7 \times 10^5$ ($\pm 9.6 \times 10^4$) gentamicin-resistant CFU from the cardia and midgut tissues, respectively (Fig 1B). Sequencing of the transformation plasmid from several bacterial clones confirmed their identity as either *Sgm*^3xant-*miR275* or *Sgm*^Scr-275. These findings indicate that rec*Sodalis* was successfully internalized by tsetse cardia and midgut cells where they were protected from the antibacterial effects of gentamicin. Additionally, significantly more rec*Sodalis* cells were present within midgut cells than cells of the cardia organ. We similarly quantified the *Sgm*^3xant-*miR275* and *Sgm*^Scr-275 present in the no gentamicin control groups (cardia, $684 \pm 90$, $p = 0.002$; midgut, $2.0 \times 10^6 \pm 1.1 \times 10^5$, $p < 0.0001$) (Fig 1B), and found that 31% and 49% of rec*Sodalis* present in the gut were intracellular within cardia and midgut tissues, respectively. These data also indicated that our rec*Sodalis* successfully reside within tsetse's gut at a density similar to that of indigenous *Sgm*^WT in age-matched flies [22]. Thus, we demonstrated that rec*Sodalis* successfully colonized tsetse's gut where they reside within cells that comprise the fly's cardia and midgut tissues.

To test the binding efficacy of the antagomirs expressed by 3xant-*miR275* to tsetse's mature *miR275*, we performed a dual luciferase reporter assay. We cloned the 3xant-*miR275* construct into the multiple cloning site located in the 3'-UTR region of the reporter gene (*renilla*) in the psiCheck-2 vector (psiCheck-2^3xant-miR275). When *miR275* binds to the sponge construct cloned in the 3'UTR region of the reporter gene (which initiates the RNA interference (RNAi) process), we expect the *renilla* transcript to be degraded, and the *renilla* Luciferase signal to be decreased. The psiCheck-2 vector also contains a *firefly* reporter in the expression cassette that is designed to be an intra-plasmid transfection normalization reporter. Thus, the *Renilla* luciferase signal is normalized to the *firefly* signal to standardize between different biological samples. We measured luciferase activity in three different experiments: 1) psiCheck-2^3xant-*miR275* + synthetic *miR275* mimic, 2) psiCheck-2^3xant-*miR275* + synthetic AllStars Negative Control, and 3) psiCheck-2^3xant-*miR275* alone, and we found that the relative luciferase activity (*renilla/firefly*) was significantly suppressed in experiment 1 compared to experiments 2 and 3 ($p < 0.05$ and $p < 0.0005$, respectively; Fig 1C). In other words, in the presence of synthetic *miR275* mimic, the luciferase activity was significantly repressed, which indicated that our sponge construct was successful when tested *in vitro* using an insect cell line. This outcome

demonstrated that the *miR275* effectively binds to the *miR275* sponge and initiates the RNAi process with its associated mRNA.

To confirm the knockdown effect of *miR275* levels *in vivo*, we used qPCR to quantify the relative expression of *miR275* in *Gmm*[3xant-*miR275*] (treatment) and *Gmm*[Scr-275] (control) individuals. Using multiple biological samples (each of which contained 5 dissected tissues pooled per sample) to reduce variability, we confirmed that the expression level of *miR275* was significantly reduced in the midgut of the treatment group compared to that of the control group ($p < 0.05$; Fig 1D). However, our qPCR results did not consistently reveal a significant reduction of *miR275* levels in the cardia organ of treatment versus control paratransgenic tsetse (Fig 1E).

## *Gmm*[3xant-*miR275*] gut physiological homeostasis is compromised

We demonstrated that rec*Sodalis* successfully invaded tsetse cardia and midgut tissues, and that *miR275* was knocked down in the midgut of *Gmm*[3xant-*miR275*]. We next sought to determine if midgut physiological processes, such as blood meal digestion and PM functional integrity, were impaired in *Gmm*[3xant-*miR275*] flies in a manner similar to what was observed when tsetse *miR275* [10] and mosquito *Ae. aegypti miR275* [16] were depleted through the use of synthetic *miR275* antagomirs. We compared the weight of midguts from 14 individual 8-day old *Gmm*[3xant-*miR275*] and *Gmm*[Scr-275] flies 24 h after their last blood meal. We observed that guts from *Gmm*[3xant-*miR275*] individuals weighed significantly more (8.37 ± 0.64 mg) than did those from *Gmm*[Scr-275] controls (4.03 ± 0.56 mg) ($p < 0.001$; Fig 2A), thus indicating that blood digestion and/or excretory processes (diuresis) were greatly disrupted in *Gmm*[3xant-*miR275*].

We next employed a highly sensitive *Serratia* infection assay to test whether PM structural integrity was compromised in paratransgenic *Gmm*[3xant-*miR275*] compared to *Gmm*[Scr-275] flies. We observed that 22% of *Gmm*[3xant-*miR275*] individuals survived for 19 days following *per os* challenge with *Serratia*. Comparatively, 0% of *Gmm*[Scr-275] control flies survived this challenge ($p < 0.0001$; Fig 2B). These data indicate that paratransgenic-mediated repression of *miR275* expression impairs tsetse's gut physiology and results in the production of a functionally

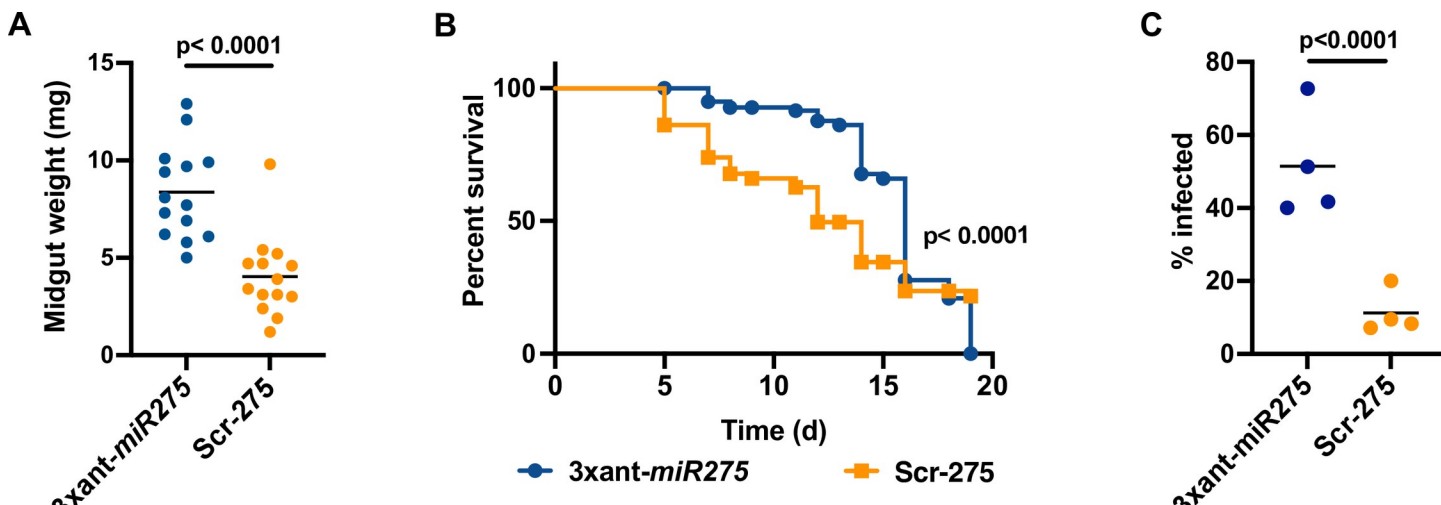

**Fig 2. Gut physiological homeostasis is compromised in *Gmm*[3xant-*miR275*].** (A) Tsetse gut weights. The gut weights were measured 24 h after the last blood meal. Each dot represents an individual fly gut. Mann-Whitney test was used for statistical analysis. (B) *Serratia* infection assay. A total of 4 biological replicates ($n = 25$ flies per replicate) were used. Gehan-Breslow-Wilcoxon test was used to determine statistical significance. (C) Trypanosome midgut infection prevalence. Four biological replicates ($n = 20$ flies per replicate) were used. Generalized linear model (GLM) with binomial distribution was used to determine statistical significance.

compromised PM barrier, similar to what we had observed using synthetic antagomirs provided *per os* in a single bloodmeal [10].

Trypanosome infection establishment success in tsetse's midgut inversely correlates with the structural integrity of the fly's PM [6,43]. We next evaluated trypanosome infection outcomes in the midgut of $Gmm^{3xant-miR275}$ relative to $Gmm^{Scr-275}$ control individuals to further confirm that paratransgenic expression of *miR275* sponges interferes with the efficacy of tsetse's PM structure. We provided 8-day old adult paratransgenic flies a blood meal containing cysteine, which inhibits trypanolytic antioxidants present in the tsetse's midgut [9,44], and $10^7$ *T. b. brucei*/mL of blood. Thereafter, the flies were maintained on normal blood meals for two weeks and subsequently dissected and microscopically examined to determine their midgut infection status. We found that significantly more $Gmm^{3xant-miR275}$ individuals (49%) hosted trypanosome infections in their gut than did their $Gmm^{Scr-275}$ counterparts (11%) ($p < 0.0001$; Fig 2C). The higher parasite infection prevalence we observed in $Gmm^{3xant-miR275}$ individuals further signifies that the functional integrity of tsetse's PM is significantly compromised when *miR275* sponges are paratransgenically expressed in the fly's midgut.

## Global gene expression profiling in paratransgenic cardia and midgut

Our paratransgenic expression system has confirmed prior phenotypes that we observed following *per os* administration of synthetic antagomir-275, including a significant reduction of *miR275* expression in the midgut and modified phenotypes associated with compromised gut physiological homeostasis such as dysfunctional digestive processes and compromised PM functional integrity. Additionally, we observed higher trypanosome infection prevalence in the midgut of $Gmm^{3xant-miR275}$ compared to $Gmm^{Scr-275}$ flies. To obtain a broader understanding of the molecular mechanisms and pathways that are regulated by *miR275*, we performed global transcriptomic profiling in cardia and midgut tissues that were harvested from paratransgenic $Gmm^{3xant-miR275}$ relative to $Gmm^{Scr-275}$ controls. All flies were age matched and inoculated *per os* with their respective rec*Sodalis* strains in their 1st and 2nd blood meals. For both comparisons each biological replicate ($n = 3$) contained pooled midguts ($n = 5$) or cardia ($n = 10$) tissues from 8-day old adults 36 h after their third blood meal. A total of 12 mRNA libraries were sequenced, and the total reads and uniquely mapped reads from each are summarized in S2 Table. We generated multi-dimensional scaling (MDS) plots to understand the overall gene expression differences between the biological replicates and treatment groups. We found that all three replicates within each treatment group clustered closely together as did all control group replicates (Fig 3A and 3B). When comparing gene expression differences in the cardia, we found that 265 genes (out of a total of 6101) were differentially expressed (DE; FDR < 0.05), with 99 (1.6%) and 166 (2.7%) up- and down-regulated in $Gmm^{3xant-miR275}$ relative to that of $Gmm^{Scr-275}$ control individuals, respectively (Fig 3A). When comparing gene expression differences in midgut samples, we found that 283 genes (out of a total of 5540) were DE (FDR < 0.05), with 116 (2.1%) and 167 (3.0%) up- and down-regulated in the midgut of $Gmm^{3xant-miR275}$ relative to $Gmm^{Scr-275}$ individuals, respectively (Fig 3B).

## Gene Ontology (GO) enrichment analysis in the paratransgenic cardia and midgut

We next applied GO enrichment analyses to acquire broad insights into the functional contributions of the DE genes we identified. In the 99 up-regulated transcripts of $Gmm^{3xant-miR275}$ cardia relative to controls, enriched GO terms included chitin binding in the molecular function category, whereas in the 166 down-regulated transcripts, enriched GO terms included iron binding, heme binding, adenosine deaminase activity, and hydrolase and peptidase

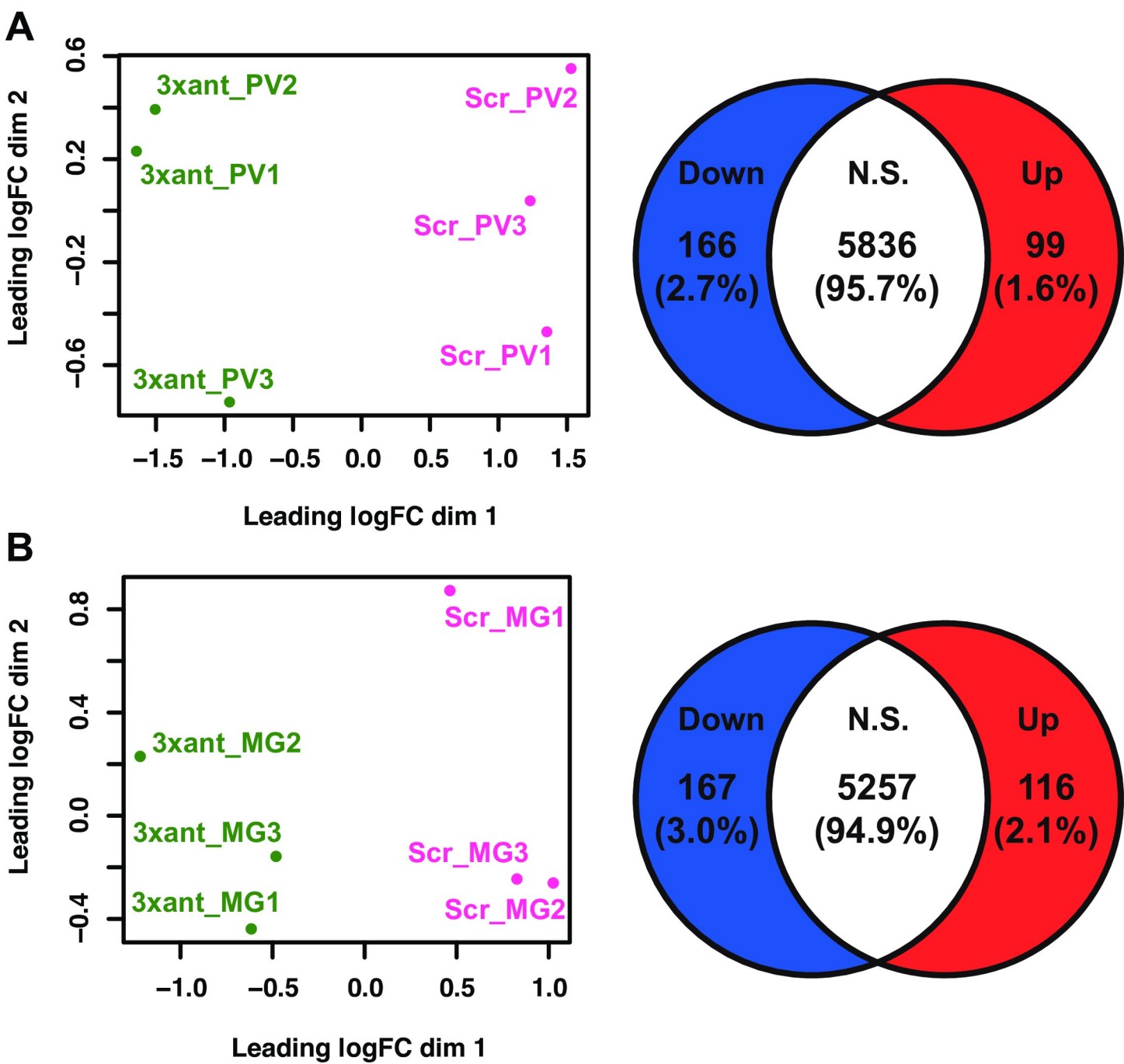

**Fig 3. Overviews of transcriptome profiles in *Gmm*<sup>3xant-*miR275*</sup> compared to *Gmm*<sup>Scr-275</sup> flies.** (A) cardia and (B) midgut transcriptome profile overview. Left panel: MDS plots display the overall gene expression patterns among the samples and between the treatments. Right panel: Venn diagrams show the number of downregulated (blue), upregulated (red) and not significantly different (white) genes in (A) cardia and (B) midgut. Genes were considered DE if they exhibited an FDR value <0.05.

activity (Fig 4A and S1 Dataset). In the 116 upregulated transcripts of *Gmm*<sup>3xant-*miR275*</sup> midguts relative to controls, enriched GO terms included catalytic activity, oxidase activity and peptidase activity in the molecular function category, while in the downregulated transcripts, enriched GO terms included ribosome and cellular component biogenesis in biological processes (Fig 4B and S1 Dataset).

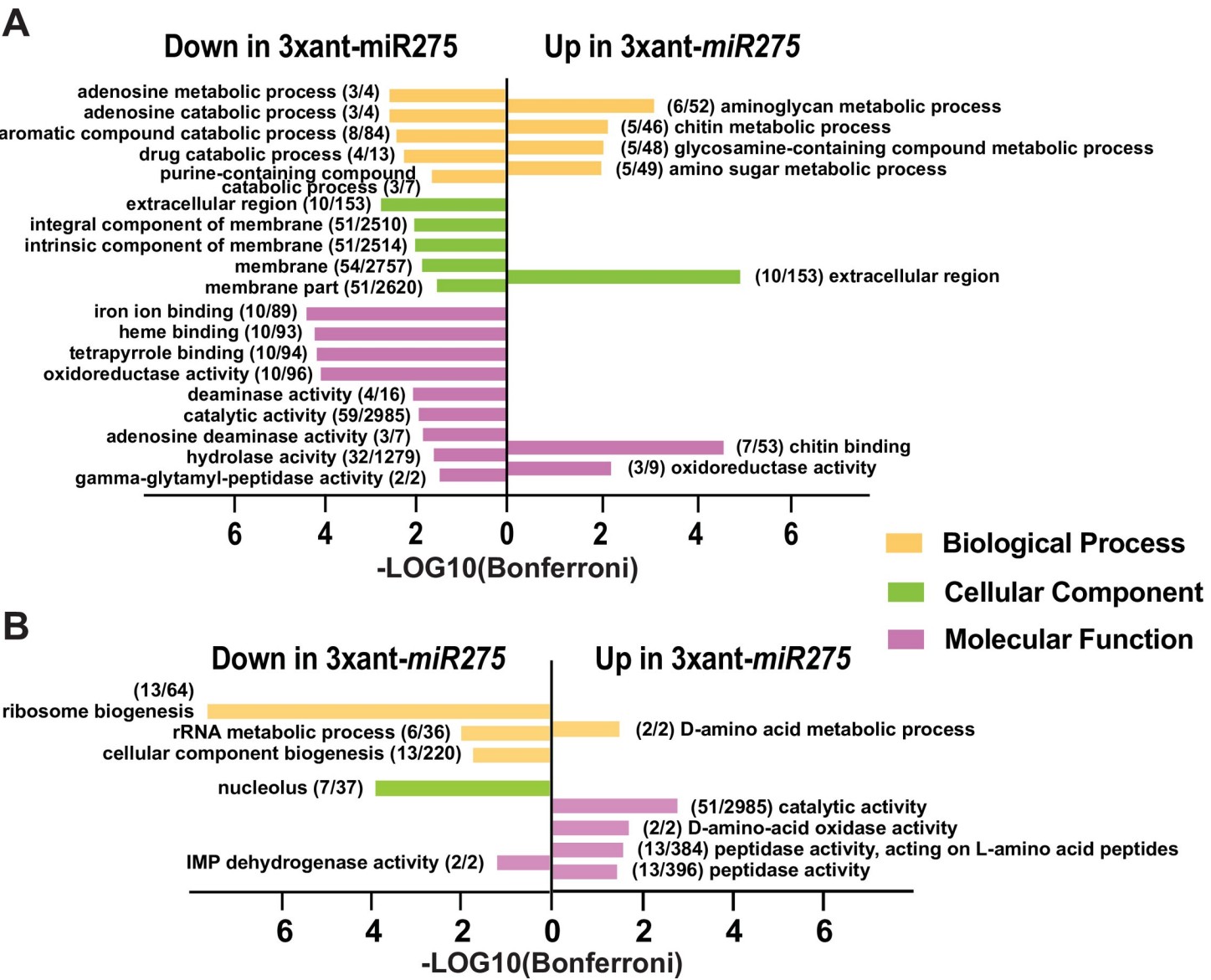

**Fig 4. GO enrichment analysis of the paratransgenic flies *Gmm*[3xant-miR275] vs. *Gmm*[Scr-275].** (A) cardia and (B) midgut tissues GO enrichment analyses. Three GO term categories were used: biological process (yellow), cellular component (green), and molecular function (pink). The GO terms were considered significant (Bonferroni score < 0.05) using VectorBase built-in GO enrichment analysis web tool. Redundant GO terms were removed by REVIGO (0.5). The number of genes in our dataset/the total number of genes that are associated to each individual GO term, are marked within parentheses next to each GO term description.

## Analysis of DE genes in the cardia from *Gmm*[3xant-miR275] vs. *Gmm*[Scr-275] control

Given that our phenotypic analysis indicated that *miR275* is involved in blood digestion and PM barrier function (Fig 2), we first evaluated the DE genes whose products are likely associated with these functions. Among the genes whose putative products have been identified as PM structural proteins through proteomics analysis of the PM [45], we found that tsetse EP, midgut trypsin (GMOY007063) and choline acyltransferase were significantly down-regulated, while serine type endopeptidase (GMOY009757), *pro1* and *GmmPer12* were up-regulated in *Gmm*[3xant-miR275] relative to *Gmm*[Scr-275] controls (Fig 5A and S2 Dataset). Among the secreted

## A. PM & digestion associated genes

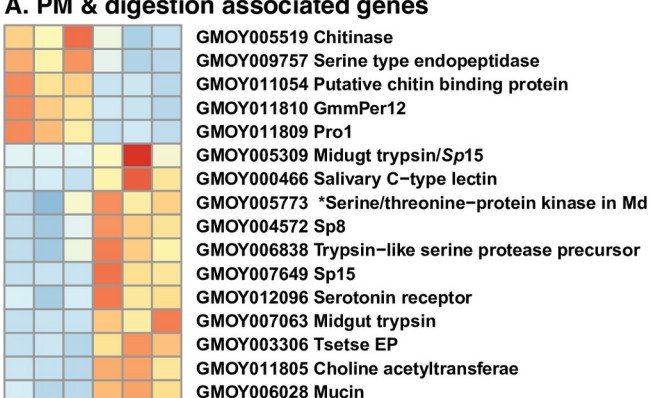

GMOY005519 Chitinase
GMOY009757 Serine type endopeptidase
GMOY011054 Putative chitin binding protein
GMOY011810 GmmPer12
GMOY011809 Pro1
GMOY005309 Midugt trypsin/*Sp*15
GMOY000466 Salivary C−type lectin
GMOY005773  *Serine/threonine−protein kinase in Md
GMOY004572 Sp8
GMOY006838 Trypsin−like serine protease precursor
GMOY007649 Sp15
GMOY012096 Serotonin receptor
GMOY007063 Midgut trypsin
GMOY003306 Tsetse EP
GMOY011805 Choline acetyltransferae
GMOY006028 Mucin
GMOY006839 Trypsin−like serine protease precursor

## C. Transporter associated genes

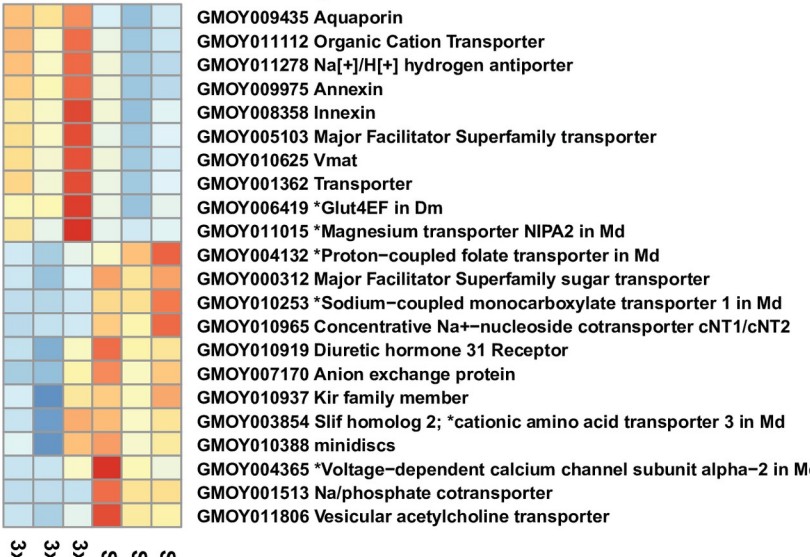

GMOY009435 Aquaporin
GMOY011112 Organic Cation Transporter
GMOY011278 Na[+]/H[+] hydrogen antiporter
GMOY009975 Annexin
GMOY008358 Innexin
GMOY005103 Major Facilitator Superfamily transporter
GMOY010625 Vmat
GMOY001362 Transporter
GMOY006419 *Glut4EF in Dm
GMOY011015 *Magnesium transporter NIPA2 in Md
GMOY004132 *Proton−coupled folate transporter in Md
GMOY000312 Major Facilitator Superfamily sugar transporter
GMOY010253 *Sodium−coupled monocarboxylate transporter 1 in Md
GMOY010965 Concentrative Na+−nucleoside cotransporter cNT1/cNT2
GMOY010919 Diuretic hormone 31 Receptor
GMOY007170 Anion exchange protein
GMOY010937 Kir family member
GMOY003854 Slif homolog 2; *cationic amino acid transporter 3 in Md
GMOY010388 minidiscs
GMOY004365 *Voltage−dependent calcium channel subunit alpha−2 in Md
GMOY001513 Na/phosphate cotransporter
GMOY011806 Vesicular acetylcholine transporter

## B. Heme binding & detoxification

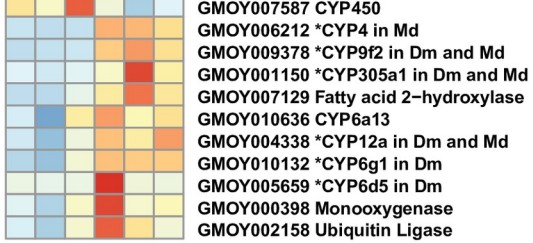

GMOY007587 CYP450
GMOY006212 *CYP4 in Md
GMOY009378 *CYP9f2 in Dm and Md
GMOY001150 *CYP305a1 in Dm and Md
GMOY007129 Fatty acid 2−hydroxylase
GMOY010636 CYP6a13
GMOY004338 *CYP12a in Dm and Md
GMOY010132 *CYP6g1 in Dm
GMOY005659 *CYP6d5 in Dm
GMOY000398 Monooxygenase
GMOY002158 Ubiquitin Ligase

## D. Saliva associated genes

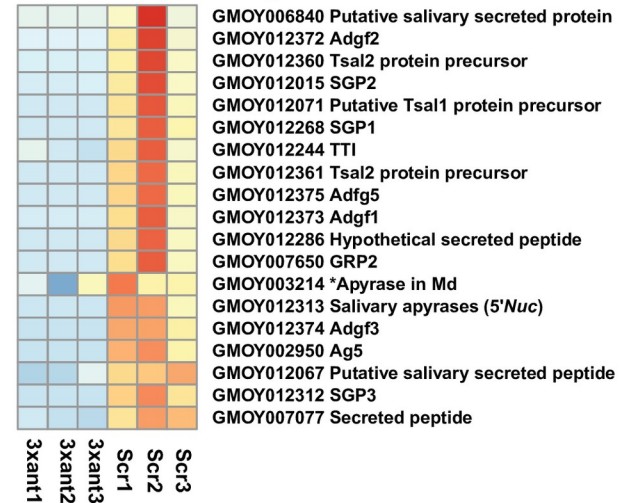

GMOY006840 Putative salivary secreted protein
GMOY012372 Adgf2
GMOY012360 Tsal2 protein precursor
GMOY012015 SGP2
GMOY012071 Putative Tsal1 protein precursor
GMOY012268 SGP1
GMOY012244 TTI
GMOY012361 Tsal2 protein precursor
GMOY012375 Adfg5
GMOY012373 Adgf1
GMOY012286 Hypothetical secreted peptide
GMOY007650 GRP2
GMOY003214 *Apyrase in Md
GMOY012313 Salivary apyrases (5'*Nuc*)
GMOY012374 Adgf3
GMOY002950 Ag5
GMOY012067 Putative salivary secreted peptide
GMOY012312 SGP3
GMOY007077 Secreted peptide

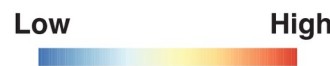

**Low** **High**

**Fig 5. Heat map representation of DE genes in different functional groups (A-D) in paratransgenic cardia *Gmm*³ˣᵃⁿᵗ⁻ᵐⁱᴿ²⁷⁵ vs. *Gmm*ˢᶜʳ⁻²⁷⁵.** (A) PM and digestion associated, (B) heme binding and detoxification, (C) transporter associated, and (D) saliva associated. Heat maps were generated by plotting the read counts in treatment (3xant-*miR275*) and control (Scr-275) samples. Colors display normalized gene expression values from low (blue) to high (red). * indicates the unknown gene product's orthologue in *Drosophila melanogaster* (*Dm*) and/or *Musca domestica* (*Md)*.

products localized to the PM, we found several digestive enzymes, serine proteases (Sp), trypsin and peptidases for which transcript abundance was significantly reduced in the treatment group (Fig 5A and S2 Dataset). The reduction in the production of these gene products may account for the impaired blood digestion we noted in *Gmm*³ˣᵃⁿᵗ⁻ᵐⁱᴿ²⁷⁵ individuals. The down-regulation of several genes whose products are associated with the PM, such as tsetse EP, midgut trypsin, *Sp* (GMOY006839), *Sp*15, and choline acyltransferase, were also noted from trypanosome-infected flies where PM functions were also compromised [9]. Tsetse EP protein is localized to the midgut, PM, and hemolymph [46,47]. The gene that encodes this protein is immune responsive, as its expression level was upregulated in response to bacterial challenge [46]. Furthermore, when tsetse EP was depleted via RNAi, trypanosome infection prevalence significantly increased [47].

Interestingly, the expression of chitinase (GMOY005519) and chitin binding protein (GMOY011054) was significantly upregulated in the cardia of $Gmm^{3xant-miR275}$ individuals. Different from other arthropod vectors, such as mosquitoes and sandflies, adult tsetse flies have type II PM, which is continuously secreted by cells located within the cardia. The PM is composed of a lattice of chitin fibrils cross linked by glycoproteins (Peritrophins) that contain chitin binding domains (CBD) [48]. Chitin is an extracellular polysaccharide that can be enzymatically hydrolyzed by chitinases [49]. Prior studies on trypanosome-infected cardia [9] and midguts [50] also indicated upregulated expression of chitinases, which likely resulted in compromised PM integrity. The reduction in PM associated gene expression, and the upregulation of the putative chitin degrading products, may contribute to the loss of PM integrity observed in paratransgenic $Gmm^{3xant-miR275}$.

With respect to blood digestion processes, we detected 10 transcripts involved in heme binding and detoxification processes that were downregulated in $Gmm^{3xant-miR275}$ compared to controls (Fig 5B and S2 Dataset). Among these putative products were cytochrome (CYP) P450 enzymes, which belong to a superfamily involved in insect metabolism, detoxification and insecticide resistance in many different species [51], as well as several CYPs regulated by Plasmodium [52] and trypanosome [53] infections. Heme in the blood can induce oxidative damage to insect tissues [54] and the presence of heme binding proteins in Ae. aegypti PM suggest the structure exhibits a detoxification role [55].

Among the transcripts encoding transporters and/or transmembrane channel proteins that would be involved in secreting, trafficking and absorbing digestive products, we detected 12 that were downregulated and 10 that were upregulated in $Gmm^{3xant-miR275}$ relative to controls (Fig 5C and S1 Dataset). These up and down-regulated genes encode functions that involve transporting nutrients such as sugar and amino acids (e.g., major facilitator super family sugar transporter, glucose transporter, Slif and minidiscs), ions and water (e.g., Na/phosphate cotransporter, calcium channel, Kir family member, magnesium transporter, and aquaporin), and organic compounds (e.g. folate transporter). Annexin and Innexin are both upregulated in $Gmm^{3xant-miR275}$. Annexin belongs to a large calcium dependent membrane binding protein family and the functions range from receptors of proteases in the gut epithelium to inhibitors of blood coagulation [56]. Plasmodium ookinetes use annexin for protection or to facilitate their development in the mosquito gut [57]. Annexin is upregulated in trypanosome-infected salivary glands (SG) [53]. Innexin proteins form gap junction channels and play critical roles in cell-to-cell communication in a variety of physiology activities [58]. Innexin 2 is a target gene of the Wingless signaling pathway in the proventricular cells in Drosophila [59]. One innexin was DE upon trypanosome infection in the tsetse fly, Glossina fuscipes fuscipes [60].

We also noted 19 abundant and significantly downregulated transcripts encoding secreted proteins in $Gmm^{3xant-miR275}$ cardia (Fig 5D and S2 Dataset), including Adenosine deaminase-related growth factor 3 (Adgf3; FC = $4.94 \times 10^{-6}$ and FDR = $1.00 \times 10^{-152}$), salivary gland protein 3 (SGP3; FC = $6.24 \times 10^{-5}$ and FDR = $1.64 \times 10^{-122}$), Antigen-5 precursor (Ag5; FC = $1.21 \times 10^{-3}$ and FDR = $2.86 \times 10^{-103}$), Tsal1 protein precursor (FC = $2.21 \times 10^{-4}$ and FDR = $1.26 \times 10^{-61}$), 5'-nucleotidase (5'Nuc; FC = $1.18 \times 10^{-4}$ and FDR = $1.10 \times 10^{-47}$), Adgf2 (FC = $2.25 \times 10^{-5}$ and FDR = $2.49 \times 10^{-36}$) and one of the two Tsal2 protein precursors (GMOY012361) (FC = $5.81 \times 10^{-5}$ and FDR = $1.86 \times 10^{-34}$) (S2 Dataset). All of these 19 genes are preferentially expressed in SG tissue and downregulated in trypanosome-infected SGs [53,61,62].

Lastly, six DE genes in $Gmm^{3xant-miR275}$ flies encoded products associated with embryogenesis and imaginal cell proliferation. Among these genes, forkhead and wing blister (Wb) were downregulated, while imaginal disc growth factor (Idgf), GMOY004790 (homologous to integrin in Md), wingless (Wg), and Wnt6 were upregulated (S2 Dataset). Idgf is involved in extracellular matrix formation in insects and participates in critical physiological activities such as

larval and adult molting and wing development [63]. The wingless pathway is an intracellular signaling network; *Wg* signaling in *Drosophila* involves embryonic epidermis and wing imaginal disc [64]. Interestingly, *Wg* expression was reduced when tsetse *miR275* was knocked down using the synthetic antagomir treatment [10], contrary to our data presented here using the constitutive silencing approach, which shows higher levels of *Wg*.

### Analysis of DE genes in the midgut from *Gmm*<sup>3xant-*miR275*</sup> vs. control *Gmm*<sup>Scr-275</sup>

Similar to our analysis with the cardia, we first analyzed DE genes that are associated with PM components and digestive enzymes in *Gmm*<sup>3xant-*miR275*</sup> midgut transcriptomes. Among previously identified PM products [45], we found 7 that were upregulated in *Gmm*<sup>3xant-*miR275*</sup> midguts, including *pro2*, *pro3*, *Sp6*, choline acetyltransferase, chitin deacetylase, midgut trypsin (GMOY007063), and a serine type endopeptidase (GMOY9757) (S3 Dataset). In addition, we also identified several digestive enzymes, including trypsin, proteases and peptidases that were upregulated in *Gmm*<sup>3xant-*miR275*</sup> midguts relative to the controls (Fig 6A and S3 Dataset). *Pro3*, *Sp6* and serine type endopeptidase (GMOY009757) were upregulated in response to *T. brucei gambiense* (*Tbg*) infection [50]. Higher levels of Chitin deacetylase, a hydrolytic enzyme that catalyzes the acetamido group in the N-acetylglucosamine units of chitin [65], could contribute to a compromised PM, similar to what we report for *chitinase* expression in the paratransgenic cardias above. The increased midgut weight we observed in *Gmm*<sup>3xant-*miR275*</sup> flies could reflect a dysfunctional gut enzyme production and/or altered enzyme transport in response to the compromised PM integrity.

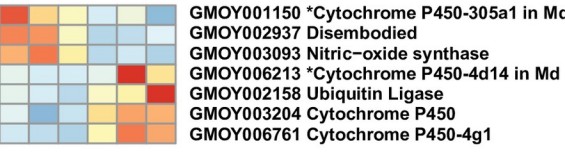

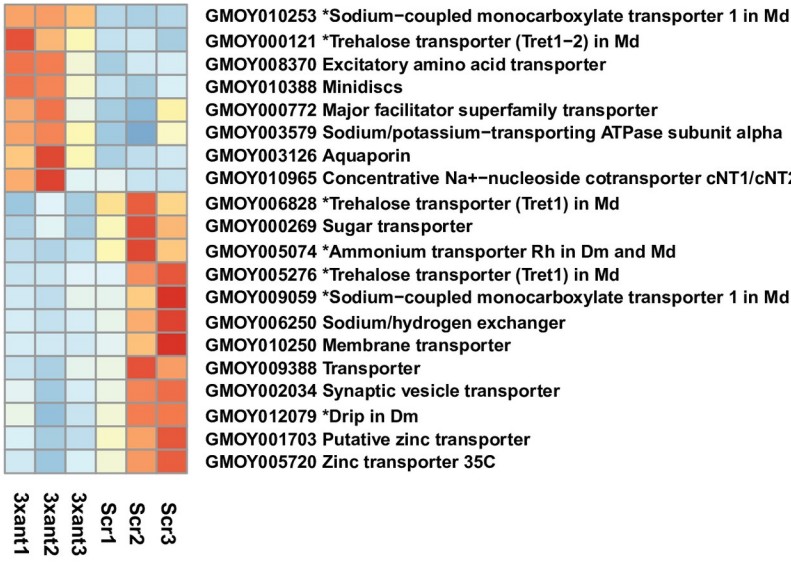

**Fig 6. Heat map representation of DE genes in different functional groups (A-C) in the midgut of *Gmm*<sup>3xant-*miR275*</sup> and *Gmm*<sup>Scr-275</sup> flies.** (A) PM and digestion associated, (B) transporter associated, and (C) heme binding and oxidative response associated. Heat maps were generated by plotting the read counts in in treatment (3xant-*miR275*) and control (Scr-275) samples. Colors display normalized gene expression values from low (blue) to high (red). * indicates the unknown gene product's orthologue in *Drosophila melanogaster* (*Dm*) and/or *Musca domestica* (*Md*).

Among the twenty genes encoding transporters and/or transmembrane channel proteins DE in the midgut (Fig 6B and S3 Dataset), two (GMOY012503 and GMOY010388) were also identified DE in the cardia of $Gmm^{3xant-miR275}$. In addition to transporters, we noted 7 DE genes, including down regulated members of $CYP$ p450, ubiquitin ligase and up regulated nitric-oxidase synthase ($NOS$) that are associated with heme binding and oxidative response (Fig 6C and S3 Dataset). The ubiquitin ligase and a heme binding protein (GMOY001150) were also down regulated in the cardia of $Gmm^{3xant-miR275}$. Ubiquitin ligase and $CYP$ p450, which are associated with insecticide resistance and metabolism of natural or xenobiotic products in many insect species [66], have been linked to toxin metabolism following a blood meal in *An. gambiae* [67]. *CYP* p450-4g1 is also DE (FC>2) in response to *Tbg* infections in the *Gmm* midgut [50]. *NOS* is responsible for producing cellular nitric oxide, which is trypanocidal [68]. *NOS* expression is down regulated in trypanosome-infected SGs [53] and cardia [9,69], and VSG-treated cardia as well [10]

Among the SG preferential genes that are dramatically reduced in $Gmm^{3xant-miR275}$ cardia, we detected five that were expressed in the midgut: *salivary C-type lectin* (GMOY000466), *Ag*5, secreted peptides (GMOY007065 and GMOY007077) and *TTI*. However, only the salivary C-type lectin was DE in the midgut and upregulated in $Gmm^{3xant-miR275}$ relative to controls.

## Paratransgenic knockdown of *miR275* is not observed in tsetse's salivary glands

We observed the significant downregulation of 19 SG preferential genes in the cardia transcriptome from $Gmm^{3xant-miR275}$ versus $Gmm^{Scr-275}$ flies. Because *per os* provisioned rec*Sodalis* is restricted in the gut tissue not in the hemolymph [22], we tested whether *miR275* is expressed in the SG (Fig 7A). We anticipated that the *miR275* knockdown effects would be restricted to the gut and not impact gene expression levels in other organs. To confirm this, we investigated whether paratransgenic knockdown of *miR275* in tsetse's gut induces a systemic response that results in the knockdown of these genes in the fly's SGs. We first dissected the SG organ from $Gmm^{3xant-miR275}$ paratransgenic flies and tested the *miR275* expression levels. We subsequently monitored the expression of *Adgf*3 (GMOY012374), *Adgf*5 (GMOY012375) and *SGP*1 (GMOY012268), which are abundantly expressed in tsetse's SGs [53,61,62] and significantly downregulated in $Gmm^{3xant-miR275}$ cardia. We found that none of the three SG-preferential genes were significantly reduced in the SG of $Gmm^{3xant-miR275}$ individuals despite being significantly down-regulated in the cardia (Fig 7B–7D). These results suggest that the effect of the paratransgenic knockdown could be restricted to tsetse's gut tissues where rec*Sodalis* reside, and is not likely to impact gene expression at the systemic level.

## Discussion

We developed a paratransgenic expression system using tsetse's endosymbiont *Sodalis* to experimentally modify *miR275* transcript abundance in tsetse's gut and to investigate the resulting physiological impact. Specifically, we engineered *Sodalis* to express *miR275* sponges (3 tandem antagomir-275 repeats), and demonstrated that the rec*Sodalis* successfully colonize tsetse's cardia and midgut where they invade resident epithelial cells. We then demonstrated that the *miR275* sponges successfully bind *miR275*, which results in posttranslational knockdown *in vitro*. We detected a significant reduction of *miR275* levels in the midgut of paratransgenic tsetse expressing *miR275* sponges, although we could not reproducibly demonstrate its reduction in the cardia organ. The paratransgenic flies displayed several robust phenotypes that are similar to those of *miR275* depletion via synthetic antagomir-275, including obstructed blood meal digestion, compromised PM functional integrity, and susceptibility to parasite

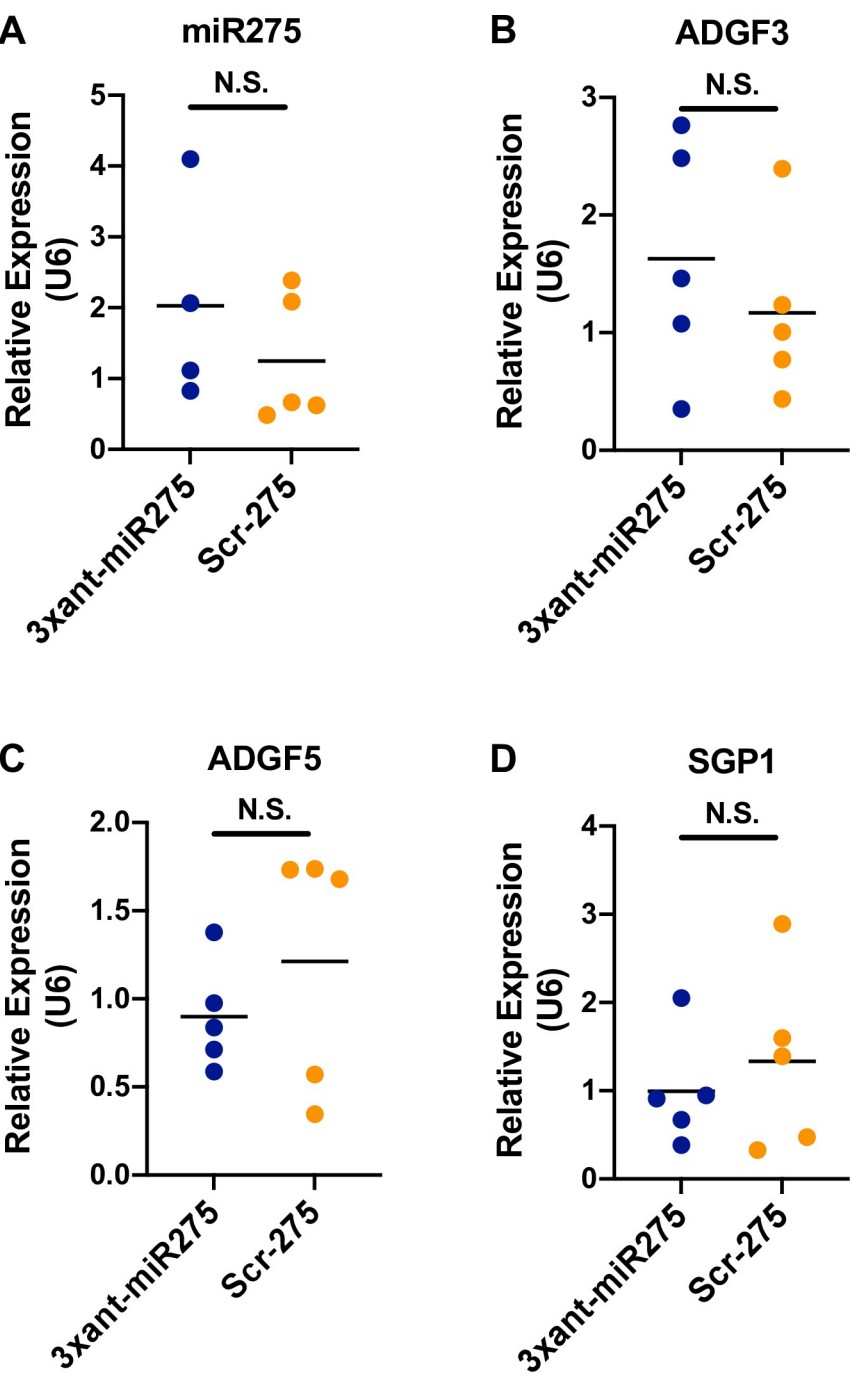

**Fig 7. Paratransgenic knockdown of tsetse *miR275* expression is not observed in the fly's salivary glands.** (A) *miR275, (B) Adgf*3, (C) *Adgf*5 and (D) *SGP*1 expression levels in the salivary glands (SGs) of *Gmm*^3xant-*miR275* versus *Gmm*^Scr-275 flies. Each dot represents 10 individual SGs. Student's t-test was used to determine statistical significance.

infection, all of which reflect impaired physiological homeostasis within the gut environment. Our transcriptomic studies further identified new molecular pathways heretofore unknown to be influenced by tsetse *miR275*, including the regulation of abundant secretory proteins functioning in vasoconstriction, platelet aggregation, coagulation, and inflammation or hemostasis. Our study is the first to use paratransgenesis as a strategy to constitutively modify the

expression of a microRNA in midgut tissue where the endosymbionts reside. It is efficient, cost effective, and minimally invasive compared to feeding and/or injecting synthetic antago-mirs, and as such, this approach serves as an efficacious alternative to investigate microRNA related functions in the tsetse fly gut. This strategy can similarly be employed in any arthropod that houses genetically modifiable commensal gut symbionts that reside within host cells [70,71].

Several experimental approaches are available to modify miRNA expression *in vivo*. Chemically synthesized, cholesterol bound antisense oligonucleotides (antagomirs) are currently most commonly used. These single stranded oligos bind their complementary endogenous miRNA, thus preventing it from interacting with its target mRNA, which inhibits downstream protein production [72]. While synthetic antagomirs interact exclusively with their complimentary miRNA, they must be administered repeatedly and often in large doses for long-term effect, their uptake by cells can be inefficient, and they are difficult to target to specific tissues [73]. Transgenic expression of miRNA sponges is another widely used method, which can provide effective and specific inhibition of miRNA seed families (the conserved sequences among miRNAs) [33]. This method, which involves the insertion of multiple, tandem antagomirs into the germline, has been successfully used to constitutively deplete miRNA abundance in mosquitoes in a tissue specific manner via the use of tissue specific promoters [17,74–76]. Because all embryonic and larval development occurs within the uterus of female tsetse [77], the generation of transgenic fly lines using traditional germline modification approaches has not been possible. To overcome this impediment, we developed the paratransgenic expression system described herein to constitutively express *miR275* sponges in tsetse's gut.

We consistently observed three phenotypes that are associated with modified tsetse midgut physiological homeostasis in our *Gmm*$^{3xant-miR275}$ flies compared to *Gmm*$^{Scr-275}$ controls. These phenotypes all correlate with the presentation of a structurally compromised PM, and they are similar to the phenotypes that we observed previously when synthetic antagomir-275 was administrated to tsetse. Specifically, we observed that *Gmm*$^{3xant-miR275}$ flies presented significantly heavier gut weights, significantly higher survival rates upon challenge with an entomopathogen, and significantly stronger vector competence, as compared to *Gmm*$^{Scr-275}$ controls. Increased midgut weight is indicative of impaired blood meal digestion and/or excretion, and this phenotype was similarly observed following treatment of *Ae. aegypti* [16] and tsetse [10] with synthetic *miR275* antagomir. In hematophagous insects, the PM mediates blood digestion by regulating the flux of digestive enzymes from their site of production in the midgut epithelium into the blood bolus-containing gut lumen [78,79]. Our study also demonstrated that significantly more *Gmm*$^{3xant-miR275}$ flies survive in the presence of entomopathogenic *Serratia* than do *Gmm*$^{Scr-275}$ control flies, further indicating that PM functional integrity is compromised in the former group of flies. *Serratia marcescens* strain Db11 is an entomopathogenic bacterium [80] that can kill tsetse when provided in the bloodmeal. Specifically, flies with an intact PM fail to immunologically detect *Serratia*, which allows the bacterium to rapidly proliferate in the gut lumen, translocate into the hemolymph and eventually to kill the tsetse and other insects [6,9,10,80–83]. Conversely, when PM structural integrity is compromised, the bacterium is quickly detected by tsetse's midgut epithelium and eliminated by the fly's robust antimicrobial immune response. The *Serratia* infection assay thus serves as a highly sensitive indicator of tsetse's PM structural integrity [6]. Lastly, we observed a higher trypanosome infection prevalence in *Gmm*$^{3xant-miR275}$ flies compared to *Gmm*$^{Scr-275}$ controls. This outcome is similar to what observed in flies exposed to anti-PM RNAi (dsRNA targeting *pro*1, *pro*2 and *chitin synthase*) [6] as well as in flies that were provisioned a blood meal containing a purified trypanosome coat protein (sVSG), which interferes with PM related gene expression in the cardia through the reduction of *miR275* [10]. Taken together, our results confirm that

interference with *miR275* expression in the cardia and midgut of *Gmm*<sup>3xant-*miR275*</sup> flies results in the modified gut environment we noted in this study.

Herein we repeatedly observed phenotypes that correspond with a depletion of *miR275* expression in tsetse's cardia. However, despite these findings, we were unable to quantify a significant reduction in expression of the microRNA in tsetse's cardia (although we could in the fly's midgut). This outcome may be accounted for by one or several reasons. First, the concentration of paratransgenically expressed *miR275* relative to the concentration of the binding sites may have reduced the inhibitory effect of the miRNA sponges [73]. Prior investigations demonstrated that tsetse *miR275* is highly abundant in the cardia compared to the midgut tissues [10]. Thus, our depletion effect could have been diluted in the cardia organ where *miR275* are highly abundant. This outcome is further exacerbated by the conspicuously low number of rec*Sodalis* that colonized cells of tsetse's cardia in comparison to the midgut. More experiments are required to optimize the uptake of rec*Sodalis* by cells of tsetse's cardia organ. Moreover, qRT-PCR can be an inaccurate method for quantifying the abundance of functional miRNAs, especially in the organ where the miRNAs are highly abundant such as tsetse's cardia. The procedure measures the total amount of miRNAs and doesn't distinguish between functional miRNAs and non-functional ones. Thus, qRT-PCR can quantify the amount of extracellular miRNA released from Trizol-lysed cells, and this represents a physiologically irrelevant population of miRNAs [84]. Combined with the robust phenotypic changes and differential expression of blood digestion and PM related genes, we believe that our paratransgenic knockdown was successful at the functional level.

Our transcriptomic analyses of cardias and midguts from paratransgenic tsetse revealed several interesting insights into the broader functions of *miR275* that are related to trypanosome infection. First, with regard to the genes that are associated with PM and digestion, midgut GO enrichment analysis indicated that downregulated genes in *Gmm*<sup>3xant-*miR275*</sup> flies included an enriched population of transcripts that encode proteins involved in ribosome biogenesis and cellular component biogenesis. This suggests that protein synthesis is obstructed in the midguts of *Gmm*<sup>3xant-*miR275*</sup> flies, which could reflect the compromised PM structure and disrupted digestion we observed in these fly's guts. GO enrichment analysis of upregulated cardia specific genes indicated that genes in *Gmm*<sup>3xant-*miR275*</sup> flies included a group of enriched transcripts that encode proteins involved in chitin metabolism and chitin binding processes. Chitinase produced by parasites degrades the sand fly and mosquito PM, which promotes *Leishmania* [85] and *Plasmodium* [86] transmission, respectively. The genome of African trypanosomes does not encode a chitinase gene. However, chitinase is a proteinaceous component of tsetse's PM, and infection with trypanosomes induces chitinase expression in the fly's cardia [9,45,87] and gut [10]. These findings suggest that parasites may facilitate their transmission through the fly by transiently upregulating cardia/gut chitinase expression, thus degrading PM chitin fibrils and reducing the structure's ability to serve as a barrier. We also observed that several genes encoding digestive enzymes were downregulated in the cardia of the *Gmm*<sup>3xant-*miR275*</sup>. Similarly, *miR275* and digestive enzyme-encoding genes (e.g., those encoding trypsin and trypsin-like proteins) were down-regulated in tsetse's cardia following trypanosome exposure [9,10]. In *Ae. aegypti*, gut-specific depletion of *miR275* results in reduced expression of its target gene *SERCA*, as well as reduced digestive enzyme secretion, disrupted gut microbiota homeostasis and compromised gut actin cytoskeleton integrity. Notably, under these circumstances, protein levels of late trypsin, a late-phase digestive protease in female mosquitoes, are significantly reduced [17]. This outcome likely accounts for the altered midgut phenotypes observed in *miR275* knockdown mosquitoes. However, tsetse *SERCA* does not contain orthologous *miR275* binding site motifs, and *SERCA* levels are not differentially expressed in *Gmm*<sup>3xant-*miR275*</sup> compared to *Gmm*<sup>Scr-275</sup> flies. These characteristics

suggest that the target of tsetse *miR275* may not be *SERCA*, and a currently unknown pathway (s) regulates the secretion of the above-mentioned proteins in tsetse's gut.

Notably, in this study the expression of some PM-associated genes was the opposite of what was observed previously when tsetse *miR275* was knocked down via synthetic antagomir-275 [10]. In the previous study, expressions of *pro*1-3 were significantly downregulated in the cardia samples after the *per os* provisioning of either synthetic antagomir-275 or sVSG. Conversely, in this study, *pro*1 in the cardia, and *pro*2 and *pro*3 in the midgut were significantly upregulated in $Gmm^{3xant\text{-}miR275}$ compared to $Gmm^{Scr\text{-}275}$ flies. However, in a different study of trypanosome-infected tsetse cardia *pro*1 is no significant different and *pro*2-3 are downregulated, and the downregulation effect of *pro* genes by provisioning sVSG in the cardia is transient [9]. This finding suggests that the *pro* genes regulation might be different based on parasite infection status. The observed disparity in *pro* gene expression by *miR275* can be explained by the possibility that the synthetic antagomir produces a one-time reduction in *miR275* expression that causes a different physiological response in the fly compared to that when *miR275* is constitutively suppressed in paratransgenic flies (rec*Sodalis* continuously produces *miR275* sponges). However, further investigation is required to acquire a more complete understanding of the *miR275* regulatory network.

Our transcriptomic results consistently showed *miR275* functions in reducing the expression of secretory enzymes and similarly impairing secretory and digestive pathways. Nineteen saliva-associated proteins were among the putative secretory products that were dramatically reduced in the cardia of the $Gmm^{3xant\text{-}miR275}$ individuals. Interestingly, seventeen of these genes were reduced in trypanosome-infected salivary glands [53,61,62], but it remains to be seen if this reduction is also mediated by lower *miR275* levels in infected salivary glands. Previous transcriptomic analyses with parasite-infected cardia indicated that 9 of these saliva protein-encoding transcripts [*Adgf*3, *Ag*5, *Tsal1*, *Tsal*2 (GMOY012360), *SGP*1, tsetse thrombin inhibitor (*TTI*), salivary secreted protein (GMOY012067) and two secreted proteins (GMOY003214 and GMOY007077)] are detected in tsetse's cardia, but only four of them [*Ag*5, *Tsal*2, *TTI* and one of the secreted proteins (GMOY007077)] are differentially expressed upon trypanosome infection [9]. Moreover, our earlier transcriptomic analysis of trypanosome-challenged tsetse guts (48 h post provisioning of a parasite containing bloodmeal) found that fourteen of these genes [*Tsal*1, *TTI*, *SGP*1, *GRP*2, 5' *Nuc*, both *Tsal*2s, *Adgf*1, *Adgf*2, *Adgf*3, *Adgf*5, salivary secreted protein and two secreted peptides (GMOY003214 and GMOY012286)] are significantly reduced relative to unchallenged controls [10]. All of these SG preferential gene products were previously detected in tsetse saliva and thought to be essential for the fly's ability to successfully blood feed [88]. In addition to being major constituents of saliva, Adgf, TTI and 5'Nuc are associated with anticoagulant functions in tsetse's saliva and gut, suggesting a role in digestive processes [60,61,89–91], while Ag5 is a major allergen involved in hypersensitivity reactions in the mammalian host [92]. The reduction of these saliva-associated anticoagulants in infected flies causes increased probing and biting behaviors, which in turn increases the transmission potential of the parasite to multiple hosts [61]. The significant reduction in expression of genes in the *Adgf* family was also very interesting. Adgf is a secreted enzyme that converts extracellular adenosine into inosine by deamination and is important in anti-inflammation, tissue damage and resistance to bacterial infection in *Drosophila* [93–95]. A Adgf is expressed by immune cells to regulate the metabolic switch during bacterial infection in *Drosophila*, and the downregulation of *Adgf* increases extracellular adenosine and enhances resistance to bacterial infection [93]. The loss of *Adgf* can induce intestinal stem cell proliferation in *Drosophila* [95]. As evidenced by reduced *Adgf* gene expressions in trypanosome-challenged tsetse guts, the downregulation of *Adgf* genes might be triggered by initial infection of trypanosomes to release anti-inflammatory response and/or to repair any

damaged tissues. Interestingly, Matetovici *et al* (2016) [53] noted significantly reduced expression of genes that encode saliva-associated products in the SGs of flies that house trypanosomes in their midgut but not yet in their SGs. This finding is suggestive of a molecular dialogue between the organs, and a possible anticipatory response of the SG environment prior to the parasites infecting the tissue, which may be mediated by *miR275* levels in these tissues. Given that these genes encode secreted proteins, their strong reduction in paratransgenic tsetse further supports the role of *miR275* in trypanosome infection, possibly through regulation of secretory pathways.

Arthropod-borne diseases impose a debilitating global public health burden. Due to the lack of effective vaccines capable of preventing the majority of these diseases, and the increasing resistance of vector arthropods to pesticides, alternative approaches for disease control are urgently needed. Paratransgenic systems have been applied in efforts to reduce vector competence in mosquitoes [29,30,96,97], kissing bugs [98,99], sand flies [100] and tsetse flies [24–26,101]. This technology has many benefits, including the absence of a reliance on inefficient germline modification procedures [85], and the fact that modified symbionts exert no fitness cost on their insect hosts [23] and can potentially spread through wild vector populations via vertical transmission [102]. Additionally, paratransgenically expressed microRNAs cost significantly less than do their synthetically produced counterparts. Our study is the first to use this system to explore the function of an arthropod vector microRNA in relation to disease transmission processes. This system can be easily applied to study the function of other tsetse miRNAs and for future research aimed at experimentally interfering with the physiological homeostasis of tsetse's midgut environment with the intent of interrupting trypanosome transmission through the fly. This study also expanded our knowledge of the relationship between tsetse *miR275* and the regulation of key physiological processes such as blood digestion, PM integrity, and gut environment homeostasis in tsetse. Our transcriptomic data revealed functions regulated by *miR275* affecting tsetse's secretory pathways. These findings provide a foundation for discovering the target of tsetse *miR275* in future studies.

## Supporting information

**S1 Table. qPCR primer list.**
(DOCX)

**S2 Table. Summary of reads mapping.**
(DOCX)

**S1 Dataset. GO enrichment analysis.**
(XLSX)

**S2 Dataset. Raw data and DE analysis of cardia transcriptome.**
(XLSX)

**S3 Dataset. Raw data and DE analysis of midgut transcriptome.**
(XLSX)

## Author Contributions

**Conceptualization:** Liu Yang, Brian L. Weiss, Emre Aksoy, Aurelien Vigneron, Serap Aksoy.

**Data curation:** Liu Yang, Brian L. Weiss, Adeline E. Williams, Alessandra de Silva Orfano, Yineng Wu.

**Formal analysis:** Liu Yang, Brian L. Weiss, Adeline E. Williams, Jae Hak Son, Serap Aksoy.

**Funding acquisition:** Serap Aksoy.

**Investigation:** Liu Yang, Brian L. Weiss, Emre Aksoy, Mehmet Karakus.

**Methodology:** Liu Yang, Brian L. Weiss, Emre Aksoy, Jae Hak Son, Serap Aksoy.

**Project administration:** Liu Yang, Brian L. Weiss, Serap Aksoy.

**Resources:** Serap Aksoy.

**Software:** Jae Hak Son.

**Supervision:** Brian L. Weiss, Serap Aksoy.

**Validation:** Liu Yang, Alessandra de Silva Orfano.

**Writing – original draft:** Liu Yang, Brian L. Weiss, Serap Aksoy.

**Writing – review & editing:** Liu Yang, Brian L. Weiss, Adeline E. Williams, Emre Aksoy, Aurelien Vigneron, Mehmet Karakus, Serap Aksoy.

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
