## [Decision Letter · Decision Letter 0]

23 Apr 2021

Dear Dr Yang

Thank you very much for submitting your manuscript "Paratransgenic manipulation of tsetse miR275 alters the physiological homeostasis of the fly’s midgut environment" for consideration at PLOS Pathogens. As with all papers reviewed by the journal, your manuscript was reviewed by members of the editorial board and by several independent reviewers. The reviewers appreciated the attention to an important topic. Based on the reviews, we are likely to accept this manuscript for publication, providing that you modify the manuscript according to the review recommendations.

Sincerely,

Marcelo Ramalho-Ortigao

Guest Editor

PLOS Pathogens

David Sacks

Section Editor

PLOS Pathogens

Kasturi Haldar

Editor-in-Chief

PLOS Pathogens

orcid.org/0000-0001-5065-158X

Michael Malim

Editor-in-Chief

PLOS Pathogens

orcid.org/0000-0002-7699-2064

Reviewer Comments (if any, and for reference):

Reviewer's Responses to Questions

**Part I - Summary**

Reviewer #1: The manuscript PPATHOGENS-D-21-00521 adds an important and exciting approach to down regulate gene expression in tsetse flies combining paratransgenesis and miRNA sponges. In addition, it adds an important contribution to understand the mechanisms used by Trypanosoma brucei parasite to manipulate its invertebrate host physiology, and further investigate the role of miR275 gene under this context.

Reviewer #2: In General, the research was well-performed, very well written, has very interesting results and the subject is of crucial importance to tsetse and the parasitic African trypanosomes control, which continues to be one the most prevalent infected disease in the world, in Africa. The authors made a complex and complete study about the manipulation of tsetse microRNA (miR275).

I have only some concerns. The authors could make the text clearer. The author summary is very well written but the abstract and introduction could be improved.

For example, on the first sentence of Abstract

“Tsetse flies are vectors of parasitic African trypanosomes (Trypanosoma spp.)” it is not necessary (Trypanosoma spp.), that could be deleted. They could also write in the abstract tsetse microRNA (miR275) instead of only "tsetse miR275 expression". In the introduction, the authors should make some modifications, adding more information about the Trypanosoma species, which is a complex genus.

Tsetse transmits two morphologically indistinguishable subspecies of the parasite that cause distinct disease patterns in humans, T. brucei gambiense and T. brucei rhodesiense. Why do not include this information? There are other Trypanosoma species very important, such as T. cruzi that causes the American trypanosomiasis, or Chagas diseases, transmitted by triatomines. Or Trypanosoma rangeli, that is also an American parasite transmitted by triatomines, more frequent than that by T. cruzi., but nonpathogenic to humans.

I suggest that the title could be changed to "Paratransgenic manipulation of a tsetse microRNA alters the physiological homeostasis of the fly’s midgut environment"

Reviewer #3: The manuscript is well written and presents a novel system for the use of transgenic bacterial symbionts to modify the expression of miRNAs in a disease vector. Previous studies have used insect symbiotic bacteria to produce dsRNA rather than miRNA antagomirs to affect gene expression, making the manuscript´s approach novel. The authors should refer to publications by Liu et al regarding the use of densovirus for a similar paratransgenic approach in Aedes albopictus. The study´s strength is the development of a novel paratransgenic system for antagomir expression and the application of omics to better understand the effects on midgut physiology of altering miRNA expression. The limitation is that there is no direct measurement of the amount of antagomirs produced by the bacteria in situ to get an estimate of the dose required for an effect. The limitation of the lack of antagomir quantification was addressed indirectly through the test of an in vitro model with insect cells to verify the efficacy of the effect. The suggestion that the amount of bacteria in the tissue can affect the level of inhibition of the miRNA needs to be verified, as the bacterial quantification and qPCR might need to be normalized for tissue weight for comparisons across tissues. For the quantification of the bacteria in midgut and cardia tissues, it might be more informative to normalize the bacterial count per mg of tissue to be able to compare between both tissues. The statistical approaches are appropriate.

**Part II - No additional experiments required**

**Part III – Minor Issues: Editorial and Data Presentation Modifications**

Reviewer #1: I have few suggestions/comments to help to improve clarity and accuracy of the manuscript and listed them below:

Line 30: miR275 is mentioned here for the first time but with no background info. Readers would greatly benefit from a short description of its importance since it is the main molecular target of this study.

Line 31: Regarding the compromising effect on the PM integrity, I suggest adding the subsequent effect on facilitating the parasite establishment in the fly gut. Although lines 344 and 345 nicely reflects this correlation, readers would greatly benefit from having this info in the abstract section.

Line 33: I suggest adding here the main hypothesis tested in this study.

Line 37: Readers would benefit from a more informative info on "modulated infection outcomes".

Line 62: Are references #2 and 3# the best choice for the disease outcome if left untreated?

Line 141: Why there are 5 nucleotides represented in black in the reverse 3xant-miR275 sequence? I would expect to find them matching with the 3-nucleotide linker sequence (atc) in the forward 3xant-miR275 sequence.

Lines 338 and 339: The percentages between control and experimental groups are switched.

Line 474 – Table S4: I suggest authors to add a new sub-sheet in Table S4 including these six genes. I could only find them in the DE sub-sheet.

Lines 518 to 521: To exclude the possibility of a systemic effect, authors should not rely on evaluating only three SG-specific genes expression in SG tissue. Other genes and tissues should not be ignored. Therefore, I must disagree with the section 3.7 subtitle (line 514) and conclusion (lines 527 to 529) stating that the effect observed is gut specific. Authors can only conclude that it does not have effect on these three SG-specific genes. The section 3.7 heading and its conclusion need careful rephrasing.

Line 545: In this context, the mentioning of "be regulated" would require additional assays, therefore I suggest using "be influenced".

Lines 577 and 578: Could authors provide references in the tsetse field that supports the statement “Increased midgut weight is indicative of impaired blood meal digestion and/or excretion”?

Typos

Line 117: Add the Gmm abbreviation after "Glossina morsitans morsitans".

Line 273: Did authors mean "newly emerged" instead of "newly eclosed"?

Line 328: I suggest "physiological processes" instead of “physiologies”.

I did not review references format throughout the whole reference list, but found names with unusual capital letters (line 829), and italics format converted into text code (945 and 946).

Reviewer #2: (No Response)

Reviewer #3: Abstract: "This paratransgenic system successfully knocked down miR275 levels in the fly’s midgut, which consequently obstructed blood digestion and modulated infection outcomes with an entomopathogenic bacteria and with trypanosomes."

It would be useful to include in the abstract by how much the expression was reduced and data on how digestion and infections were affected.

Methods:

It would be useful to clarify in the methods section that in the Serratia model the bacteria will not infect if the PM is compromised.

p. 22 line 406 State the full name of the EP gene.

p. 29 Line 552 It would be useful for the reader to include some specific examples of similar models where bacteria could be genetically modified.

p. 33 Line 659 With the current data, it is not proven that the antagomir is constitutively produced, only one time point was evaluated.

References: all species should be in italics, and some references have a different format (ex. line 885). The paratransgenic approach using densovirus (Lie et al) is a relevant reference to include.

Line 994 Correct student´s to Student´s t test

Figure 7 "gut tissue specific" should be reworded, as other tissues were not tested. It might be better to state that the system does not affect expression in salivary glands at the tested timepoint. Effects at other time points or tissues were not tested.

There are several paragraphs in the results section that should be in the discussion section. For ex.

p. 26 line 490 "Higher levels..."

p. 25 line 465 "All of these..."

p. 24 line 461 "Morover..."

p. 23 Line 435 "These up and downregulated..."

Figure 1. cfu/midgut does not normalize for the size of the tissues. For the reader to better appreciate the differences in bacterial loads between the two tissues, it would be better to normalize using tissue mg as denominator.

Figure 1 D and E. Is it possible that no difference was observed in expression in the cardia because the tissue amount was too low? The legend indicates that 5 cardia and 5 midguts were used, but these tissues most probably have different sizes that could affect the efficiency of the qPCR method if it is not normalized by mg of tissue. p. 12 Line 210 "A small portion of the RNA" may not allow for comparison across tissues. This should be addressed as a possible limitation or potentially include previous optimization results for the qPCR to show that this variability in input RNA does not affect the efficiency of the qPCR.

Figure Files:

Data Requirements:

Reproducibility:

References:

---

## [Editor Report · Decision Letter 1]

13 May 2021

Dear Dr. Yang,

We are pleased to inform you that your manuscript 'Paratransgenic manipulation of a tsetse microRNA alters the physiological homeostasis of the fly’s midgut environment' has been provisionally accepted for publication in PLOS Pathogens.

Best regards,

Marcelo Ortigao

Guest Editor

David Sacks

Section Editor

PLOS Pathogens

Kasturi Haldar

Editor-in-Chief

PLOS Pathogens

orcid.org/0000-0001-5065-158X

Michael Malim

Editor-in-Chief

PLOS Pathogens

orcid.org/0000-0002-7699-2064
---

## [Editor Report · Acceptance letter]

7 Jun 2021

Dear Dr. Yang,

We are delighted to inform you that your manuscript, "Paratransgenic manipulation of a tsetse microRNA alters the physiological homeostasis of the fly’s midgut environment," has been formally accepted for publication in PLOS Pathogens.

Best regards,

Kasturi Haldar

Editor-in-Chief

PLOS Pathogens

orcid.org/0000-0001-5065-158X

Michael Malim

Editor-in-Chief

PLOS Pathogens

orcid.org/0000-0002-7699-2064